# Indene-Derived Hydrazides Targeting Acetylcholinesterase Enzyme in Alzheimer’s: Design, Synthesis, and Biological Evaluation

**DOI:** 10.3390/pharmaceutics15010094

**Published:** 2022-12-28

**Authors:** Shraddha Manish Gupta, Ashok Behera, Neetesh K. Jain, Devendra Kumar, Avanish Tripathi, Shailesh Mani Tripathi, Somdutt Mujwar, Jeevan Patra, Arvind Negi

**Affiliations:** 1Faculty of Pharmacy, Oriental University, Indore 453555, India; 2Department of Pharmaceutical Sciences, School of Health Sciences and Technology, University of Petroleum and Energy Studies (UPES), Dehradun 248007, India; 3Faculty of Pharmacy, DIT University, Dehradun 248009, India; 4Institute of Pharmaceutical Research, GLA University, Mathura 281406, India; 5School of Chemical Sciences and Pharmacy, Central University of Rajasthan, Ajmer 305817, India; 6Chitkara College of Pharmacy, Chitkara University, Rajpura 140401, India; 7Department of Bioproduct and Biosystems, Aalto University, FI-00076 Espoo, Finland

**Keywords:** indene-hydrazide conjugates, fused ring scaffolds, hyrazide conjugation, acetylcholinesterase, Alzheimer’s disease, ADMET screening, SH-SY5Y cell line, donepezil

## Abstract

As acetylcholinesterase (AChE) plays a crucial role in advancing Alzheimer’s disease (AD), its inhibition is a promising approach for treating AD. Sulindac is an NSAID of the aryl alkanoic acid class, consisting of a indene moiety, which showed neuroprotective behavior in recent studies. In this study, newer Indene analogs were synthesized and evaluated for their in vitro AChE inhibition. Additionally, compared with donepezil as the standard drug, these Indene analogs were accessed for their cell line-based toxicity study on SH-SY5Y cell line. The molecule **SD-30**, having hydrogen bond donor (HBD) at para-position, showed maximum AChE inhibition potential (IC_50_ 13.86 ± 0.163 µM) in the indene series. Further, the **SD-30** showed maximum BuChE inhibition potential (IC_50_ = 48.55 ± 0.136 µM) with a selectivity ratio of 3.50 and reasonable antioxidant properties compared to ascorbic acid (using DPPH assay). **SD-30** (at a dose level: of 10 µM, 20 µM) effectively inhibited AChE-induced Aβ aggregation and showed no significant toxicity up to 30 mM against SH-SY5Y cell lines.

## 1. Introduction

The synthesis of fused aromatic ring scaffolds as a pharmacophore has immensely interested organic medicinal chemists in targeting intracellular and extracellular proteins [1,2]. The conjugation of the fused ring scaffolds with other aromatic systems or polyaromatic systems is commonly practiced by the medicinal chemist to bring photochemical control [3], improving physicochemical parameters [4] and reducing on-target toxicity [5]. Although fused ring scaffolds are widely implemented in various diseases, their implementation in neurodegenerative disease is still lower because of their limited physicochemical properties. One such neurodegenerative disease is Alzheimer’s dementia (AD), which is clinically manifested with lowered levels of acetylcholine. Acetylcholine is a neurotransmitter; any abruption in its levels also leads to other diseases, such as Lambert-Eaton myasthenic syndrome (LEMS) and myasthenia gravis (MG). Numerous neurons, namely forebrain cholinergic neurons, become damaged due to the formation of serine hydrolase enzymes, which is linked to aberrant acetylcholine metabolism and impaired IQ. Based on epidemiology studies, 44.4 million individuals worldwide had dementia as of 2013, and it is expected to reach 75.6 million globally by 2030 and 135.5 million worldwide by 2050, especially in developing nations. Approximately 62% of people who have dementia reside in third-world countries, which will become 71 percent by 2050 [6]. Although the essential etiological factor for AD has not yet been identified, the hallmarks of this disorder are recognized abnormal brain diseases, such as the over-expression of Acetylcholinesterase (AChE) and the extracellular deposition of “mystery”-amyloid plaques. In the brain, the cholinergic neurons engage in cortical activity, cerebral blood flow, memory- and learning-oriented processes, and the modulation of cognition [7]. The choline acetyltransferase enzyme (AChE) is depleted explicitly in the brains of AD patients, which lowers acetylcholine synthesis and affects cortical cholinergic functions [7]. Therefore, inhibiting the cholinesterase enzymatic activity in the synaptic cleft was found to be a promising strategy to improve AD symptoms. Butyrylcholinesterase (BChE) is the other cholinesterase (ChE), which makes 20% of the total brain ChE activity. Clinical studies showed elevated levels of BChE during AD onset, making it a potential target in AD treatment [8]. Unlike AChE, BChE is not expressed in the peripheral and autonomous nervous systems; therefore, selective BChE inhibitors would be devoid of the adverse side effects as what is commonly seen with AChE-specific inhibitors, which would reduce the threshold dosing. In the context of AD, cymserine analogue selective targeting of BChE alleviated the secretion of amyloid β (Aβ) peptide in a human neuronal cell line, and enhances cognitive performance in aged rats [9].

To date, there is no cure for AD, and numerous reversible AChE inhibitors are in clinical trials. One of the major challenges that the drug designer commonly faces during AD drug development is to find a suitable molecular scaffold that can comply with the distinctive cellular physiochemical properties (specifically, the blood-brain barrier) in the brain. Therefore, molecules with higher AChE potency do not warrant their higher blood-brain barrier (BBB) crossing. However, in some instances, an increase in dosage showed a marginal enhancement in BBB crossing but was flawed in producing on-target toxicities on smooth muscles. Therefore, the rational designing of molecular structure targeting AChE should display preferential cholinesterase selectivity to mitigate the on-target toxicity in the peripheral cellular system and typical physicochemical characteristic (ADMET) to cross BBB. To an extent, donepezil, rivastigmine, and galantamine were clinically developed as cholinesterase inhibitors (as shown in Figure 1A), which enhance cholinergic transmission and provide symptomatic relief for a portion of dementia patients [10]. Interestingly, donepezil is an indene-1-one analog, which shows preferential selectivity towards AChE than BChE. However, indene scaffolds are also devised to develop Mcl-1 inhibitors [11], estrogen-targeting compounds [12,13], and naturally derived fungal metabolites [12], as shown in Figure 1B.

In our previously reported study, we performed virtual screening on 44 indene analogs, using flexible molecular docking to identify the binding mechanism of these indene-containing ligands to AChE [14]. First, the resulting indene analogs (as hits) were synthesized, followed by biochemical assays performed (AChE-induced A1-42 aggregation assay, in vitro AChE, and BuChE inhibition assays, the antioxidant activity DPPH assay, a cell line-based toxicity investigation on the SH-SY5Y cell line).

## 2. Materials and Methods

### 2.1. General Materials and Methods

All chemicals, reagents, and solvents for the synthesis of indene analogs were purchased from commercial sources and were used without further purification. Indene analogs were synthesized starting from sulindac using a slightly modified protocol as previously described by Bhat et al. [15]. The reactions were monitored using thin-layer chromatography on silica gel aluminum sheets and visualized with a UV lamp. NMR spectra were recorded on Bruker AV 600 MHz and 300 MHz spectrometers, operating at 150.92 or 75.47 MHz for ^13^C and 600.13 or 300.13 MHz for ^1^H nuclei. Chemical shifts are quoted in ppm and are referenced to SiMe4 as an internal standard unless stated otherwise. Multiplets are abbreviated as follows: br—broad; s—singlet; d—doublet; t—triplet; q—quartet; m—multiplet. Mass spectra (ESI-MS) were obtained on Shimadzu LCMS 2010-A spectrometer, Japan. Compounds were purified using column chromatography with silica gel as the stationary phase and ethyl acetate/Petroleum acetate mixtures (5:5) as the eluent system.

### 2.2. General Procedure for the Synthesis of Indene Ester

A mixture consisting of 1 g (0.0028 mol) of sulindac, 11.89 g (15 mL, 2.7 mol) of absolute methanol, and 10 drops of concentrated sulphuric acid was placed in a round-bottomed flask with a capacity of 100 mL. After attaching a reflux condenser, the mixture was heated over low heat for a period of five hours. TLC was used to monitor the reaction in different polarity of binary solvent systems using ethyl acetate (EA) and petroleum ether (PE) (EA:PE = 20, 50, and 80%), respectively. The reaction solvent was removed under reduced pressure. The solid residue was dissolved in ethyl acetate and washed with fresh water using a 100 mL separating funnel. The mixture was then vigorously shaken in the separating funnel to separate the components of the mixture. After being allowed to stand, two distinct layers emerged and were collected individually. The organic layer (ethyl acetate layer) was collected, the remaining water layer was poured back into the separating funnel, and fresh ethyl acetate was added. The collected ethyl acetate fraction was washed with fresh water, then dried over 5 g of anhydrous magnesium/sodium sulfate. This process was repeated until no further effervescences of carbon dioxide were observed. The flask was given vigorous shaking for approximately five minutes before being left to rest for at least half an hour while being shaken intermittently. Following filtration of the ethyl acetate solution with a small, fluted filter paper directly into an RBF, the solution was then subjected to rotary evaporation to remove the solvent. After collecting the sulindac methyl ester as a yellow, viscous liquid, the methyl ester was then recrystallized from ethyl ether to give a yellow solid in quantitative yield [16].

### 2.3. General Procedure for the Synthesis of Hydrazide Analogs of Indene

Sulindac methyl ester (0.01 mol) and hydrazine hydrate (0.2 mol) were refluxed in ethanol (50 mL) for 30 h. TLC was used to monitor the reaction, and the reaction solvent (ethanol) was removed under reduced pressure. Later, the semi-solid residue was dissolved in crushed ice with continuous stirring, and the precipitates were filtered and washed with fresh water. These precipitates were collected and crystallized in ethanol, which yielded yellow colored solid with Yield (70%) and m.p. 120–122 °C.

### 2.4. General Procedure for the Synthesis of Indene Derivatives

A solution of Sulindac hydrazide (1 mmol) in ethanol (15 mL) containing appropriately substituted benzaldehydes (1.1 mmol) and a catalytic amount of glacial CH_3_COOH was heated under reflux for 3 h. The reaction mixture was added to ice-cold water in a beaker. The product was precipitated, filtered by vacuum filtration, and washed several times with cold water. The solid was recrystallized from ethanol (NMR spectra are shown in Appendix A).

*2-(5-fluoro-2-methyl-1-((E)-4-(methylsulfinyl) benzylidene)-1H-inden-3-yl)-N’-((E)-3-hydroxy benzylidene)acetohydrazide* (**SD-24**): **SD-24** was synthesized using 3-hydroxybenzaldehyde (**Comp. no 50**) (0.13 g) as pale yellow compound, yield 60%; mp 240–240 °C; FTIR (KBr) cm^−1^: 3470–3400 (-OH, -NH), 3030–3070 (Ar), 1680 (C=O), 1070 (SO); ^1^H-NMR (500 MHz, DMSO-d_6_) δ 8.31 (s, 1H, acetohydrazide -NH), 7.79 (s, 1H, benzylidene -CH), 7.61 (d, 2H, (methylsulfinyl)benzylidene-C_2_, C_6_), 7.56 (d, 2H, (methylsulfinyl)benzylidene-C_3_, C_5_), 7.41 (s, 1H, inden-C_4_), 7.35 (s, 1H, 3-hydroxy benzylidene -CH), 7.19 (t, 1H, 3-hydroxy benzylidene-C_5_), 7.09–6.98 (m, 2H, inden-C_6_, C_7_), 6.95–6.82 (m, 3H, 3-hydroxy benzylidene-C_2_, C_2_, C_6_), 3.66 (s, 1H, 3-hydroxy benzylidene -OH), 3.10 (s, 2H, acetohydrazide -CH_2_), 2.41 (s, 3H, (methylsulfinyl)benzylidene-CH_3_), 1.71 (s, 3H, inden-CH_3_); ^13^C-NMR (125 MHz, DMSO-d_6_) δ 166.15, 161.74, 161.51, 156.78, 147.92, 145.14, 142.09, 139.58, 137.46, 137.00, 131.94, 130.70, 129.87, 129.87, 129.50, 129.48, 126.53, 126.53, 122.18, 120.00, 119.36, 114.73, 112.66, 103.84, 43.49, 36.09, 14.59; MS (ESI): *m*/*z* found 475.149 [M^+^]; calcd. 475.15. [M^+^ + 1]; calcd. 476.14.

*2-(5-fluoro-2-methyl-1-((E)-4-(methylsulfinyl) benzylidene)-1H-inden-3-yl)-N’-((E)-3-methoxy benzylidene) acetohydrazide* (**SD-25**): **SD-25** was synthesized using 3-methoxybenzaldehyde (**Comp. no 51**) (0.14 g) as cream colored compound, yield 65%; mp 130–132 °C; FTIR (KBr) cm^−1^: 3386–3400 (-NH), 3019–3080 (Ar), 1677 (C=O), 1066 (SO); 1H-NMR (500 MHz, DMSO-d6) δ 8.29 (s, 1H, acetohydrazide -NH), 7.65 (s, 1H, benzylidene -CH), 7.59 (d, 2H, (methylsulfinyl)benzylidene-C2, C6), 7.52 (d, 2H, (methylsulfinyl)benzylidene-C3, C5), 7.38 (s, 1H, inden-C4), 7.32 (s, 1H, 3-hydroxy benzylidene -CH), 7.16 (t, 1H, 3-hydroxy benzylidene-C5), 7.07–6.96 (m, 2H, inden-C6, C7), 6.92–6.78 (m, 3H, 3-hydroxy benzylidene-C2, C2, C6), 3.07 (s, 2H, acetohydrazide -CH2), 2.38 (s, 3H, (methylsulfinyl)benzylidene-CH3), 2.33 (s, 3H, 3-methoxy benzylidene -CH3), 1.66 (s, 3H, inden-CH3); 13C-NMR (125 MHz, DMSO-d6) δ 166.15, 161.74, 161.51, 160.29, 147.92, 145.14, 142.09, 139.58, 137.46, 137.02, 131.94, 129.87, 129.87, 129.50, 129.48, 129.20, 126.53, 126.53, 122.18, 121.23, 114.89, 112.66, 112.19, 103.84, 56.04, 43.49, 36.09, 14.59; MS (ESI): *m*/*z* found 488.23 [M^+^]; calcd. 488.16 [M^+^ + 1]; calcd. 489.16.

*2-(5-fluoro-2-methyl-1-((E)-4-(methylsulfinyl)benzylidene)-1H-inden-3-yl)-N’-((E)-4-hydroxy benzylidene)acetohydrazide* (**SD-30**): **SD-30** was synthesized using 4-hydroxybenzaldehyde (**Comp. no 52**) (0.13 g) as pale yellow colored compound, yield 50%; mp 210–212 °C; FTIR (KBr) cm^−1^: 3391–3410 (-OH, -NH), 3024–3085 (Ar), 1681 (C=O), 1072 (SO); ^1^H-NMR (500 MHz, DMSO-d_6_) δ 8.18 (s, 1H, acetohydrazide -NH), 7.63 (s, 1H, benzylidene -CH), 7.61 (d, 2H, (methylsulfinyl)benzylidene-C_2_, C_6_), 7.55 (d, 2H, (methylsulfinyl)benzylidene-C_3_, C_5_), 7.40 (s, 1H, inden-C_4_), 7.35 (d, 2H, 4-hydroxy benzylidene -C_3_, C_5_), 7.20 (d, 2H, 4-hydroxy benzylidene-C_2_, C_6_), 7.09–6.99 (m, 2H, inden-C_6_, C_7_), 3.71 (s, 1H, 4-hydroxy benzylidene -OH), 3.10 (s, 2H, acetohydrazide -CH_2_), 2.41 (s, 3H, (methylsulfinyl)benzylidene-CH_3_), 2.35 (s, 3H, 3-methoxy benzylidene -CH_3_), 1.68 (s, 3H, inden-CH_3_); ^13^C-NMR (125 MHz, DMSO-d_6_) 166.15, 161.74, 161.51, 158.32, 148.64, 145.14, 142.09, 139.58, 137.46, 131.94, 129.87, 129.87, 129.72, 129.72, 129.50, 129.48, 126.92, 126.53, 126.53, 122.18, 115.60, 115.60, 112.66, 103.84, 43.49, 36.09, 14.59; MS (ESI): *m*/*z* found 475.149 [M^+^]; calcd. 475.55. [M^+^ + 1]; calcd. 476.152.

*2-(5-fluoro-2-methyl-1-((E)-4-(methylsulfinyl)benzylidene)-1H-inden-3-yl)-N’-((E)-2-methoxy benzylidene) acetohydrazide* (**SD-31**): **SD-31** was synthesized using 2- methoxybenzaldehyde (**Comp. no 53**) (0.14 g) as yellow colored compound, yield 55%; mp 173–176 °C; FTIR (KBr) cm^−1^: 3379–3398 (-NH), 3015–3087 (Ar), 1669 (C=O), 1068 (SO); ^1^H-NMR (500 MHz, DMSO-d_6_) δ 8.23 (s, 1H, acetohydrazide -NH), 7.59 (s, 1H, benzylidene -CH), 7.54 (d, 2H, (methylsulfinyl)benzylidene-C_2_, C_6_), 7.50 (d, 2H, (methylsulfinyl)benzylidene-C_3_, C_5_), 7.36 (s, 1H, inden-C_4_), 7.29 (s, 1H, 3-hydroxy benzylidene -CH), 7.13 (t, 1H, 3-hydroxy benzylidene-C_5_), 7.04–6.94 (m, 2H, inden-C_6_, C_7_), 6.89–6.74 (m, 4H, 2-methoxy benzylidene-C_3_, C_4_, C_5_, C_6_), 3.05 (s, 2H, acetohydrazide -CH_2_), 2.33 (s, 3H, (methylsulfinyl)benzylidene-CH_3_), 2.31 (s, 3H, 2-methoxy benzylidene -CH_3_), 1.58 (s, 3H, inden-CH_3_); ^13^C-NMR (125 MHz, DMSO-d_6_) δ 166.15, 161.74, 161.51, 159.58, 149.83, 145.14, 142.09, 139.58, 137.46, 131.94, 129.87, 129.87, 129.50, 129.48, 128.33, 127.89, 126.53, 126.53, 125.51, 122.18, 121.54, 113.69, 112.66, 103.84, 56.79, 43.49, 36.09, 14.59; MS (ESI): *m*/*z* found 488.18 [M^+^]; calcd. 488.58 [M^+^ + 1]; calcd. 489.58.

*2-(5-fluoro-2-methyl-1-((E)-4-(methylsulfinyl) benzylidene)-1H-inden-3-yl)-N’-((E)-2-methoxy benzylidene) acetohydrazide* (**SD-40**): **SD-40** was synthesized using 2,6-dihydroxybenzaldehyde (**Comp. no 54**) (0.14 g) to get compound as creamish solid compound, yield 55%; mp 182–182 °C; FTIR (KBr) cm^−1^: 3500–3411 (-OH, -NH), 3020–3093 (Ar), 1666 (C=O), 1051 (SO); ^1^H-NMR (500 MHz, DMSO-d_6_) δ 8.10 (s, 1H, acetohydrazide -NH), 7.48 (s, 1H, benzylidene -CH), 7.46 (d, 2H, (methylsulfinyl)benzylidene-C_2_, C_6_), 7.41 (d, 2H, (methylsulfinyl)benzylidene-C_3_, C_5_), 7.31 (s, 1H, inden-C_4_), 7.22 (s, 1H, 2,6-dihydroxybenzylidene -CH), 7.00–6.88 (m, 2H, inden-C_6_, C_7_), 6.81–6.73 (m, 3H, 2,6-dihydroxybenzylidene-C_3_, C_4_, C_5_), 4.25 (s, 2H, 2,6-dihydroxy benzylidene -OH), 2.88 (s, 2H, acetohydrazide -CH_2_), 2.21 (s, 3H, (methylsulfinyl)benzylidene-CH_3_), 1.41 (s, 3H, inden-CH_3_); ^13^C-NMR (125 MHz, DMSO-d_6_) δ 166.15, 161.74, 161.51, 159.69, 159.69, 145.14, 143.26, 142.09, 139.58, 137.46, 131.98, 131.94, 129.87, 129.87, 129.50, 129.48, 126.53, 126.53, 122.18, 112.66, 112.56, 109.25, 109.25, 103.84, 43.49, 36.09, 14.59; MS (ESI): *m*/*z* found 490.13 [M^+^]; calcd. 490.55, [M^+^ + 1]; calcd. 491.55.

*2,3-dihydroxybenzylidene)-2-(5-fluoro-2-methyl-1-((E)-4-(methylsulfinyl) benzylidene)-1H-inden-3-yl)acetohydrazide* (**SD-42**): **SD-42** was synthesized using 2,3-dihydroxybenzaldehyde (Comp. no 55) (0.14 g) to get creamish compound, yield 52%; mp 220–222 °C; FTIR (KBr) cm^−1^: 3508–3406 (-OH, -NH), 3013–3084 (Ar), 1702 (C=O), 1081 (SO); ^1^H-NMR (500 MHz, DMSO-d_6_) δ 8.24 (s, 1H, acetohydrazide -NH), 7.58 (s, 1H, benzylidene -CH), 7.51 (d, 2H, (methylsulfinyl)benzylidene-C_2_, C_6_), 7.45 (d, 2H, (methylsulfinyl)benzylidene-C_3_, C_5_), 7.37 (s, 1H, inden-C_4_), 7.27 (s, 1H, 2,3-dihydroxybenzylidene -CH), 7.09–6.95 (m, 2H, inden-C_6_, C_7_), 6.90–6.80 (m, 3H, 2,6-dimethoxy benzylidene-C_4_, C_5_, C_6_), 4.33 (s, 2H, 2,6-dihydroxy benzylidene -OH), 2.93 (s, 2H, acetohydrazide -CH_2_), 2.24 (s, 3H, (methylsulfinyl)benzylidene-CH_3_), 1.45 (s, 3H, inden-CH_3_); ^13^C-NMR (125 MHz, DMSO-d_6_) δ 166.15, 161.74, 161.51, 149.42, 146.95, 145.14, 144.69, 142.09, 139.58, 137.46, 131.94, 129.87, 129.87, 129.50, 129.48, 126.53, 126.53, 122.18, 122.01, 121.33, 119.94, 119.85, 112.66, 103.84, 43.49, 36.09, 14.59; MS (ESI): *m*/*z* found 490.23 [M^+^]; calcd. 490.55, [M^+^ + 1]; calcd.491.55.

### 2.5. Cholineseterase Inhibition

Enzyme activities were determined using the Ellman spectrophotometric method [17,18]. Enzyme substrates acetylthiocholine iodide (ATCh) and propionylthiocholine iodide (PTCh) were purchased from Sigma-Aldrich, Steinheim, Germany, and thiole reagent 5.5′-dithiobis-2-nitrobenzoic acid (DTNB) from Sigma-Aldrich, St. Louis, MO, USA. Donpezil and were used as reference compounds (purchased from sigma);

In initial state, ATCh (derived from human erythrocytes) and PTCh (derived from human plasma) were dissolved in water and DTNB in 0.1 M sodium phosphate buffer (pH 7.4). Sulindac analogues were dissolved in water, and all further dilutions were made in water. Final concentrations of synthetics were in the range of 0.01–200 µM, while substrates were 1.0 mM and 4.0 mM for ATCh and PTCh, respectively. The final dilution of AChE and BChE was 500 and 300 times, respectively. Briefly, 50 µL of AChE (1.00 U mL^−1^) or 50 µL of BuChE (0.6 U mL^−1^) and 20 µL of the test or standard compounds were incubated in 96 well plates for 30 min at room temperature. Further, 100 µL (1.5 mM) of DTNB was added to the above solution. The substrate i.e., ATCI (15 mM, 10 µL) or BTCI (30 mM, 10 µL) was added to it, and absorbance was recorded immediately at 415 nm for 20 min at 1 min interval using Synergy HTX multi-mode reader (BioTek, USA) [19,20]. The IC_50_ values were calculated using absorbance obtained from the test and standard compounds. The assays were performed in triplicate with three independent runs.

### 2.6. Cell Line-Based Toxicity Study

The toxicity of compound **SD-30** was evaluated against SH-SY5Y neuroblastoma cell lines by the MTT assay according to the literature procedure with minor modifications [20]. Briefly, the cell lines (density 1 × 10^5^ cells/wells) were plated in 96 well plates and incubated for 24 h at 37 °C in CO2 incubator. Various concentrations of **SD-30** (10, 20,40, and 50 mM) were added, and cells were incubated for 72 h. After that, 20 mL of MTT reagent was added, and the cells were incubated for an additional 2 h. The obtained purple colored formazan crystals were solubilized in 100 mL DMSO. The absorbance was measured at 570 nm, and the % cell viability was calculated. Each treatment was executed in triplicate and data are presented as percentage of the control.

### 2.7. Enzyme Induced Aβ 1-42 Aggregation Assay

Thioflavin T (ThT) assay was performed to determine the inhibition potential of **SD-30** against AChE induced Aβ 1-42 aggregation. Aβ 1-42, purchased from Sigma, dissolved in the 10 mM phosphate buffer (PBS) of pH 7.5 to get a stock concentration of 200 mM. The test compounds were dissolved in distilled water and further diluted with PBS. Different proportions of the Aβ 1-42: Inhibitor (1:0.5, 1:10, 1:20) were used in the ThT assay. The final concentration of Aβ 1-42, **SD-30** and AChE was 10 μM (2 μL), 0.5, 10, 20 μM (2 μL) and 230 μM (16 μL) respectively. The mixtures were incubated at room temperature for 48 h in the dark. The fluorescence intensities of the incubated mixtures were measured by adding 178 μL of 20 μM ThT at excitation and emission wavelengths of 485 and 528 nm at the end of the experiment. The ThT assay was performed in triplicate and in three independent runs.

### 2.8. Antioxidant Assay

The assay was performed by previously mentioned protocol [19,21]. Briefly, 10 and 20 μM (10 μL) of the synthesized compounds (Tris-HCl buffer, pH 7.4) were mixed with 20 μL, 10 mM (stock in methanol) of DPPH (Hi-Media) in 96 well plate. The volume of solution was adjusted to 200 μL using methanol. The 96-well plate was incubated at 37 °C for 25 min on shaking water bath with moderate shaking. The absorbance of reaction mixture was measured at 520 nm wavelength. The reducing percentage (RP) of the DPPH was determined by the equation:$$\mathrm{RP}=100\;\frac{\left(A_{0}-A_{c}\right)}{A_{0}}$$
where A_0_ was the control or untreated DPPH absorbance, and A_c_ was the test treated DPPH absorbance. Ascorbic acid was used as the standard for the DPPH assay. The assay was performed in triplicate.

### 2.9. Statical Analysis

Data were analyzed by GraphPad Prism 9.4.0.673. One Way ANOVA with multiple comparisons was used in the cell toxicity determination. One-way ANOVA followed by one-way analysis of variance *** *p* < 0.0001 and standard deviation (SD) was calculated in the antioxidant study.

## 3. Results and Discussion

### 3.1. Design and Synthesis of Compounds

The ligand library with 44 methylated indene derivatives was designed by substituting various functionalities with diverse electronic features like electron donor and acceptor functional groups at positions R_1_ and R_2_. The compounds which showed potent AChE binding affinity were synthesized (as shown in Figure 1) through a three-step process, which includes esterification, the nucleophilic addition of hydrazide derivatives, and imine-coupling with aromatic aldehydes [22]. Later, the reaction solvent was evaporated under reduced pressure, and the resulting solid residue was then purified by column chromatography. Finally, FTIR, ^1^H-NMR, ^13^C-NMR, and HRMS spectral techniques were used for molecular characterization.

### 3.2. AChE and BuChE Inhibition Assay

The inhibition potency of the synthesized indene analogs were evaluated for both BchE and AchE. All compounds have exhibited a time-dependent inhibition on both BchE and AchE, demonstrating first-order kinetics (Table 1).

Since AD is a complex disease regulated via multiple factors, we have developed multi-targeting molecules to counter the disease. The first goal was to assess the effect of hydroxy and methoxy groups on the in vitro enzyme inhibition. These groups were substituted at the benzylidene *ortho*/*meta* and *para*-positions. The involvement of hydrogen bonds in the drug interaction with the macromolecular targets was discovered by inserting hydrogen-bond donor (HBD) groups at the *meta*/*para*-position of the indene series of compounds. Molecule **SD-30** having HBD at para position, showed maximum AChE inhibition potential (IC_50_ = 13.86 ± 0.163 µM) in the indene series. Further, the same molecule showed maximum BuChE inhibition potential (IC_50_ = 48.55 ± 0.136). Furthermore, the molecule **SD-30** was found to be selective for AChE (3.50). Shifting the position of the OH group to the *meta*-position (**SD-24**, *h*AChE IC_50_ = 40.43 ± 0.067 µM; *h*BuChE IC_50_ = 92.86 ± 0.066 µM) leads to a decrease in inhibition potential on both targets. Further, the effect of the di-hydroxy group was assessed. Two hydroxy groups containing compounds **SD-40** (2,6-OH) and **SD-42** (2,3-OH) showed furthermore decrease in inhibition potential on *h*AChE and *h*BuChE (hAChE IC_50_ = 108.86 ± 0.142 µM, hBuChE IC_50_ = 286.01 ± 0.143 µM and hAChE IC_50_ = 92.44 ± 0.106 µM, *h*BuChE IC_50_ = 264.33 ± 0.068 µM respectively). However, the substitution of the benzylidene ring with a methoxy group at ortho and meta positions worsen the activity (**SD-25**: *h*AChE IC_50_ > 1000, hBuChE IC_50_ > 1000; **SD-31**: *h*AChE IC_50_ 733.16 ± 0.143 µM, *h*BuChE IC_50_ > 1000).

### 3.3. Evaluation of Cell Line-Based Toxicity Study on SH-SY5Y Cell Line

Generally, in vitro cell line studies have been routinely used as these can be helpful in primary optimization and mechanistic investigations before animal-based toxicology studies [23]. The evaluation of the neuroprotection effect of compound **SD-30** (10, 20, 40, and 50 mM) on human neuroblastoma SH-SY5Y cell lines was assessed through MTT (3-(4,5-dimethylthiazol-2-yl)-2,5- diphenyl tetrazolium bromide) assay [23,24] (as shown in Figure 2). This cell line is often used in AD due to its human origin and ease of maintenance [25].

### 3.4. AChE-Induced Aβ1-42 Aggregation Assay

AChE enzyme is reported to facilitate the aggregation of Aβ. Thus, the potent compound **SD-30** was evaluated for its ability to inhibit AChE induced Aβ aggregation [26,27]. Aβ_1-42_ and AChE treated Aβ_1-42_ are 100% aggregated after the incubation of 48 h. The marketed compound Donapezil showed a significant decrease in the Aβ_1-42_ aggregation at 10 and 20 µM. Further, the compound **SD-30** was also found to be effective on aggregation at doses of 10 and 20 µM (Figure 3).

## 4. Conclusions

From our previous study, based on MEP analysis, geometric optimization and molecular reactivity analysis showed stability as **SD-24** > **SD-30** = **SD-42**. 2-(5-fluoro-2-methyl-1-{(E)-4-(-(methyl sulfinyl) benzylidene}-1H-inden-3-yl)-N’-(substituted benzylidene) acetohydrazide derivatives were synthesized in high yields. The most potent compound (**SD-30**) showed potent inhibition for AChE (IC_50_ = 13.86 ± 0.163 µM) and for BuChE (IC_50_ = 48.55 ± 0.136 µM). Furthermore, compounds **SD-24**, **SD-30**, and **SD-42** showed a reasonable antioxidant property, with potency **SD-42** > **SD-30** ≥ **SD-24**. The neuroprotective effect was accessed using MTT assay against human neuroblastoma SH-SY5Y cell lines, where cytotoxicity of **SD-30** was not observed up to 30 mM. Furthermore, **SD-30** was also found to be effective on Aβ_1-42_ aggregation at doses of 10 and 20 µM. Based on these biochemical studies, **SD-30** was found to be a promising compound that can be further explored for AD treatment.

## Data Availability

The data presented in this study are available on request from the corresponding author (Arvind Negi).

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
