# Peer review of "Indene-Derived Hydrazides Targeting Acetylcholinesterase Enzyme in Alzheimer’s: Design, Synthesis, and Biological Evaluation"

_pharmaceutics, 2022, doi:10.3390/pharmaceutics15010094_

Round 1
Reviewer 1 Report
This research work is in need for extensive improvements:
1-Figure 1, the selected drugs didnot supported the choice of indene as a scaffold, it means poor rationale, has to be improved.
2-Scheme 1 starts with compound 47 which isnot acceptable as compounds no. have to be arranged sequencialy all over the manuscript.
3-Additionally R substituents are not listed.
4-What is the differnces between compounds 50 to 55 ?
5-The same to scheme 2, compounds are not given any numbers and R are not mentioned.
6-Table 1, where are these numbers in the manuscript, why compounds are distributed and not arranged, why you have choose these compounds for screening?
7-Figure 6 with bad resolution.
8-please attach the spectra of the new compounds as supplementary data.
Author Response
We would like to thank the reviewer for taking out time and providing comments.
The changes were highlighted in yellow color with track changes on for convenience.
1-Figure 1, the selected drugs didnot supported the choice of indene as a scaffold, it means poor rationale has to be improved.
Author's Response: We added a section in the introduction accordingly.
2-Scheme 1 starts with compound 47 which isnot acceptable as compounds no. have to be arranged sequencialy all over the manuscript.
Author's Response: Scheme 1 is redrawn. However, a virtual screening was initially performed where the naming of compounds starts with #1, 2, 3, 4, 5….. (that can be found in supplementary information), and hits were considered for the synthesis.
3-Additionally R substituents are not listed.
Author's Response: It is now listed in scheme 1.
4-What is the differences between compounds 50 to 55 ?
Author's Response: Revised accordingly.
5-The same to scheme 2, compounds are not given any numbers and R are not mentioned.
Author's Response: Scheme 2 is now removed.
6-Table 1, where are these numbers in the manuscript, why compounds are distributed and not arranged, why you have choose these compounds for screening?
Authors Response: Table 1 is now revised, and the numbering is corrected.
The Lead Compounds' Selection Criteria: The final hits were chosen using two sets of selection principles that spanned two phases of the process and are based on binding affinity, drug-likeness properties, and physicochemical interactions between the ligand and the target protein. In the phase one selection criteria of blind docking, docking scores less than -8 Kcal/mol were used to screen ligands. In the second phase, three parameters were taken into account when selecting additional potential hits: redocking scores that were less than or equal to those of the co-crystal ligand docking scores using the appropriate docking tools, the ability to interact with amino acids involved in acceptor substrate positioning, and drug likeliness properties as determined by Lipinski's rule of five, Veber's rule parameter, Muegge's rule, and toxicology studies. These screening criteria enabled us to identify potential hits for MDSs, DFT, and rational AD drug design.
7-Figure 6 with bad resolution.
Author's Response: Revised accordingly.
8-please attach the spectra of the new compounds as supplementary data.
Author's Response: The new compound proton and carbon spectral data are now included in SI.
Reviewer 2 Report
Recommendation: Reconsider after major revision.
The present manuscript entitled “Indene-derived hydrazides targeting Acetylcholinesterase enzyme in Alzheimer: Design, Synthesis, and Biological Evaluation” by Shraddha Manish Gupta and et. al. synthesized Indene analogs and evaluated their in-vitro acetyltransferase (AChE), Butyrylcholinesterase (BuChE) inhibitions, followed by a cell line-based toxicity study on SH-SY5Y cell line. The authors reported a selective Indene analog which exhibited IC50 value 13.86 ± 0.163 and 48.55 ± 0.136 μM against AChE and BuChE respectively. The authors also performed docking and MD simulation studies to understand the binding mode of the selective inhibitor against AChE. This is an interesting work; however, the following points should be addressed before publication can be considered:
Ø In the introduction section, the authors discussed only AChE enzyme. A brief discussion about BuChE is needed in the introduction as well. Since, both the proteins were considered for bioassay experiments.
Ø On page 6 at the “2.6 Docking Studies” section, the authors gave the PDBID of BuChE but not for AChE enzyme. The authors should provide the PDBID of AChE as well in this section.
Ø How many Indene analogs were chosen for docking studies?
Ø It is not clear how the authors define the active site of BuChE and AChE? Whether they considered center-of-mass of the active site residues of BuChE for grid generation?
Ø On page 6 at line number 257-259, the authors mentioned that they shortlisted the docked poses based on their maximum binding score which were computed by Biovia Discovery Studio. However, the authors performed docking using Autodock Vina which is a different package than Biovia. The authors should clarify this. Which score was finally used for ranking best docking poses.
Ø The authors should replace the section 3.1. “Design and Synthesis of Compounds” in the method.
Ø What force field parameters were used for protein and ligand during MD simulation? The authors did not mention which ensemble (say, NVT or, NPT etc.) was used during equilibration and production MD simulation?
Ø In the abstract, the authors mentioned that they compared their study with the donepezil. However, no comparison is given between IC50 values of donepezil and the Indene derivatives. The authors should perform this study and report in this manuscript.

Author Response
We would like to thank the reviewer for taking out time and providing comments.
The changes were highlighted in yellow color with track changes on for convenience.
The present manuscript entitled “Indene-derived hydrazides targeting Acetylcholinesterase enzyme in Alzheimer: Design, Synthesis, and Biological Evaluation” by Shraddha Manish Gupta and et. al. synthesized Indene analogs and evaluated their in-vitro acetyltransferase (AChE), Butyrylcholinesterase (BuChE) inhibitions, followed by a cell line-based toxicity study on SH-SY5Y cell line. The authors reported a selective Indene analog which exhibited IC50 value 13.86 ± 0.163 and 48.55 ± 0.136 μM against AChE and BuChE respectively. The authors also performed docking and MD simulation studies to understand the binding mode of the selective inhibitor against AChE. This is an interesting work; however, the following points should be addressed before publication can be considered:
- In the introduction section, the authors discussed only AChE enzyme. A brief discussion about BuChE is needed in the introduction as well. Since, both the proteins were considered for bioassay experiments.
Authors Response: A section dedicated in the introduction, which includes the therapeutic significance of BuChE in AD treatment.
- On page 6 at the “2.6 Docking Studies” section, the authors gave the PDBID of BuChE but not for AChE enzyme. The authors should provide the PDBID of AChE as well in this section.
Authors Response: Thank you for your close observation. The pdb id for the AChE enzyme was mentioned in the manuscript.
- How many Indene analogs were chosen for docking studies?
Authors Response: 44 indene analogs were screened in an earlier stage, and the data is provided in the supplementary information.
- It is not clear how the authors define the active site of BuChE and AChE? Whether they considered center-of-mass of the active site residues of BuChE for grid generation?
Authors Response: Thank you for your valuable suggestion. The active site for both the macromolecular targets (BuChE and AChE) were generated by placing the grid centre with respect to the centre point of the cocrystallized ligand and by extending it to cover the interacting residues of the macromolecular target to cover the extended conformations of the ligand.
- On page 6 at line number 257-259, the authors mentioned that they shortlisted the docked poses based on their maximum binding score which were computed by Biovia Discovery Studio. However, the authors performed docking using Autodock Vina which is a different package than The authors should clarify this. Which score was finally used for ranking best docking poses.
Authors Response: Thank you for your close observation. The Autodock vina was used for performing docking experiments, while docking poses were retrieved with the help of Biovia Discovery Studio visualizer tool.
- The authors should replace the section 3.1. “Design and Synthesis of Compounds” in the method.
Authors Response: It is replaced now.
- What force field parameters were used for protein and ligand during MD simulation? The authors did not mention which ensemble (say, NVT or, NPT etc.) was used during equilibration and production MD simulation?
Authors Response: Charmm36-jul2021 force field is used for evaluate the stability of protein-ligand complex during molecular dynamics simulations. Equilibration was performed for 100ns in NVT and NPT ensemble using V-rescale thermostat and Parrinello-rahman barostat for 300K at 1bar pressure. Finally the whole system was subjected to an MD production run for 100ns using NPT.
- In the abstract, the authors mentioned that they compared their study with the donepezil. However, no comparison is given between IC50 values of donepezil and the Indene derivatives. The authors should perform this study and report in this manuscript.
Authors Response: IC50 values of donepezil are mention in Table1
Reviewer 3 Report
The authors of the manuscript titled "Indene-derived hydrazides targeting Acetylcholinesterase enzyme in Alzheimer: Design, Synthesis, and Biological Evaluation" report the Synthesis of newer Indene analogs, evaluated for their in-vitro AChE inhibition, followed by cell line-based toxicity study on SH-SY5Y cell line, compared with donepezil as the standard drug. The body of work presented here is appropriate for Pharmaceutics; however, it needs some major revisions before it can be considered for publication.
Points that need to be addressed.
- In section 3.1, the authors discuss a library of 44 methylated indene derivatives designed for docking analysis. However, only 6 compounds were synthesized out of the 44, which is a statistically too small pool of compounds to make any inferences from the biological data. There needs to be a proper SAR, need a more extensive library of compounds for biological assays to make a proper conclusion regarding the activities of these scaffolds. Additionally, there was no justification for synthesizing only six compounds out of 44.
- The authors need to be consistent with numbering as scheme 1 has numbers from 47 to 55, and the methods section and table 1 have the SD numbers. Make sure the numbering system is consistent.
- The authors must provide all the H NMR and C NMR spectra in the supporting information. Additionally, please provide Table S1 in the supporting information. There is a need for an entire supporting information section.
- There is no need for scheme 2 as it is just a repetition of scheme 1. For scheme 1 please provide all the R information along with the yields. Correct the numbers (please use the subscripts) for all the reagents and solvents in the scheme legend.
- The authors need to provide a better-quality figure 6 for the manuscript. Please enlarge it and increase the quality of figure 6.
- For Table 1, please provide positive controls donepezil and tacrine etc., for the IC50 inhibitory activities against AChE and BuChE.
- For figures 2, 3, and 4, Please increase the size of the panels for better quality. Also, where are the structure of compounds 30 and 42?
- There were no statistical analyses of data. The authors should add a section in the methods on statistical analyses and carry out appropriate tests in order to support their inferences. Please include them in methods or provide them in the figure legends.
- The authors have provided the data for the inhibition of Ab1-42 oligomer assembly. However, no studies were done regarding the dissociation of the preformed Ab1-42 oligomer.
- Mechanism of AChE/BuChE inhibition. Do the authors have any input on whether the compounds inhibited AChE and BuChE by the same mechanism?
- For section 2.5, the authors need to provide the details of inhibition studies. Did the authors incubate the test compounds and enzymes for some time before adding substrate and DTNB? In that case, did the authors check for the possibility of progressive covalent inhibition?
Author Response
We would like to thank the reviewer for taking out time and providing comments.
The changes were highlighted in yellow color with track changes on for convenience.
The authors of the manuscript titled "Indene-derived hydrazides targeting Acetylcholinesterase enzyme in Alzheimer: Design, Synthesis, and Biological Evaluation" report the Synthesis of newer Indene analogs, evaluated for their in-vitro AChE inhibition, followed by cell line-based toxicity study on SH-SY5Y cell line, compared with donepezil as the standard drug. The body of work presented here is appropriate for Pharmaceutics; however, it needs some major revisions before it can be considered for publication.
Points that need to be addressed.
- In section 3.1, the authors discuss a library of 44 methylated indene derivatives designed for docking analysis. However, only 6 compounds were synthesized out of the 44, which is a statistically too small pool of compounds to make any inferences from the biological data. There needs to be a proper SAR, need a more extensive library of compounds for biological assays to make a proper conclusion regarding the activities of these scaffolds. Additionally, there was no justification for synthesizing only six compounds out of 44.
Response 1: Authors are thankful for the valuable suggestions of the reviewer. As indene derivatives reported for Alzheimer (Alexander A. Titov et. al., ACS Chem. Neurosci. 2021, 12, 2, 340-353, Mehmet Koca et. al., Journal of enzyme inhibition and medicinal chemistry. 2016 Nov 2;31(sup2):13-23.), therefore we initiated our study with a virtual screening of 44 compounds, filtered with their docking energy and ADMET data (included in supporting information). Those hits (or compounds) that showed high docking scores, and drug-likeness properties, were further selected for Synthesis. However, due to limited resources, we only focused on the Synthesis and biochemical characterisation for 6 compounds.
- The authors need to be consistent with numbering as scheme 1 has numbers from 47 to 55, and the methods section and table 1 have the SD numbers. Make sure the numbering system is consistent.
Response 2: The authors have highly apologized for the inconvenience. Now, the numbering is consistent in the edited manuscript.
- The authors must provide all the H NMR and C NMR spectra in the supporting information. Additionally, please provide Table S1 in the supporting information. There is a need for an entire supporting information section.
Response 3: Supporting information is included.
- There is no need for scheme 2 as it is just a repetition of scheme 1. For scheme 1 please provide all the R information along with the yields. Correct the numbers (please use the subscripts) for all the reagents and solvents in the scheme legend.
Response 4: Scheme 1 has been edited and completely redrawn again , while scheme-2 has been removed from the revised manuscript.
- The authors need to provide a better-quality figure 6 for the manuscript. Please enlarge it and increase the quality of figure 6.
Response 5: Figure 6 has been edited.
- For Table 1, please provide positive controls donepezil and tacrine etc., for the IC50 inhibitory activities against AChE and BuChE.
Response 6: AChE and BuChE IC50 of positive control donepezil are included in the modified manuscript (Table 1).
- For figures 2, 3, and 4, Please increase the size of the panels for better quality. Also, where are the structure of compounds 30 and 42?
Response 7: revised accordingly. The strictures of 30 and 42 is in scheme 1
- There were no statistical analyses of data. The authors should add a section in the methods on statistical analyses and carry out appropriate tests in order to support their inferences. Please include them in methods or provide them in the figure legends.
Response 8: Section 2.9 (2.9. Statical analysis) is added in the material and method section. The method of statical analysis is also mentioned in the figure legend (figure 5, 6).
- The authors have provided the data for the inhibition of Ab1-42 oligomer assembly. However, no studies were done regarding the dissociation of the preformed Ab1-42 oligomer.
Response 9: Authors are thankful for the suggestion to include the dissociation of the preformed Ab1-42 oligomer study in the manuscript. Currently, our lab is not equipped for such studies. We will surely include such characterization in our future communication.
- Mechanism of AChE/BuChE inhibition. Do the authors have any input on whether the compounds inhibited AChE and BuChE by the same mechanism?
Response 10: Detail mechanism study of AChE and BuChE inhibition is beyond the scope of the current manuscript. The result reported in the current manuscript may be taken forward for the study of the detailed mechanism.
- For section 2.5, the authors need to provide the details of inhibition studies. Did the authors incubate the test compounds and enzymes for some time before adding substrate and DTNB? In that case, did the authors check for the possibility of progressive covalent inhibition?
Response 11: The detailed procedure is added in the modified manuscript (Section 2.5).
The test compounds were incubated for 30 min. with the enzyme. Finally, substrates and
DTNB were mixed, and immediate absorbance was recorded. Assessment of the possibility of progressive covalent and non-covalent inhibition is an important study to find out the kinetics of drug-enzyme interaction. Our lab will collaborate with the experts in this field and include such studies in our future communications.
Round 2
Reviewer 1 Report
It can be accepted now.
Author Response
Thanks for the comments.
We also revised the manuscript further to improve the sentences in the main manuscript.

Reviewer 2 Report
The authors have addressed my comments satisfactorily. The only comment I would like to pass on to the authors regarding the incomplete figure-1 legend. The authors must describe figure 1A and 1B in the legend.
Author Response
Thanks for the comment, and we add legends for Figure 1.
We also revised the manuscript further to improve the sentences in the main manuscript.
